

# Fall risk prediction ability in rehabilitation professionals: structural equation modeling using time pressure test data for Kiken-Yochi Training

Ryohei Kishita[1,2], Hideki Miyaguchi[1], Tomoko Ohura[3], Katsuhiko Arihisa[4], Wataru Matsushita[5] and Chinami Ishizuki[1]

[1] Graduate School of Biomedical & Health Sciences, Hiroshima University, Hiroshima, Hiroshima, Japan
[2] Department of Occupational Therapy, Faculty of Health Sciences, Osaka University of Human Sciences, Settsu, Osaka, Japan
[3] Center for Gerontology and Social Science, Research Institute, National Center for Geriatrics and Gerontology, Obu, Aichi, Japan
[4] Division of Occupational Therapy, Department of Rehabilitation Sciences, Faculty of Allied Health Sciences, Kansai University of Welfare Sciences, Kashiwara, Osaka, Japan
[5] Department of Occupational Therapy, School of Health Sciences at Fukuoka, International University of Health and Welfare, Okawa, Fukuoka, Japan

Corresponding author
Hideki Miyaguchi, hmiya@hiroshima-u.ac.jp

## ABSTRACT

**Background**. Falls occur frequently during rehabilitation for people with disabilities. Fall risk prediction ability (FRPA) is necessary to prevent falls and provide safe, high-quality programs. In Japan, Kiken Yochi Training (KYT) has been introduced to provide training to improve this ability. Time Pressure-KYT (TP-KYT) is an FRPA measurement specific to fall risks faced by rehabilitation professionals. However, it is unclear which FRPA factors are measured by the TP-KYT; as this score reflects clinical experience, a model can be hypothesized where differences between rehabilitation professionals (licensed) and students (not licensed) can be measured by this tool.
**Aims**. To identify the FRPA factors included in the TP-KYT and verify the FRPA factor model based the participants' license status.
**Methods**. A total of 402 participants, with 184 rehabilitation professionals (physical and occupational therapists) working in 12 medical facilities and three nursing homes, and 218 rehabilitation students (physical and occupational therapy students) from two schools participated in this study. Participant characteristics (age, gender, job role, and years of experience and education) and TP-KYT scores were collected. The 24 TP-KYT items were qualitatively analyzed using an inductive approach based on content, and FRPA factors were extracted. Next, the correction score (acquisition score/full score: 0–1) was calculated for each extracted factor, and an observation variable for the job role (rehabilitation professional = 1, rehabilitation student = 0) was set. To verify the FRPA factors associated with having or not having a rehabilitation professional license, FRPA as a latent variable and the correction score of factors as an observed variable were set, and structural equation modeling was performed by drawing a path from the job role to FRPA.
**Results**. The results of the qualitative analysis aggregated patient ability (PA), physical environment (PE), and human environment (HE) as factors. The standardized coefficients of the model for participants with or without a rehabilitation professional

license and FRPA were 0.85 ($p < 0.001$) for FRPA from job role, 0.58 for PA, 0.64 for PE, and 0.46 for HE from FRPA to each factor ($p < 0.001$). The model showed a good fit, with root mean square error of approximation $< 0.001$, goodness of fit index (GFI) $= 0.998$, and adjusted GFI $= 0.990$.

**Conclusion**. Of the three factors, PA and PE were common components of clinical practice guidelines for fall risk assessment, while HE was a distinctive component. The model's goodness of fit, which comprised three FRPA factors based on whether participants did or did not have rehabilitation professional licenses, was good. The system suggested that rehabilitation professionals had a higher FRPA than students, comprising three factors. To provide safe and high-quality rehabilitation for patients, professional training to increase FRPA should incorporate the three factors into program content.

# INTRODUCTION

Safe activities of daily living are essential for maintaining and improving the health of people with disabilities (*Sheth et al., 2019*; *World Health Organization, 2023*). Rehabilitation uses physical, occupational, and speech-language therapies to achieve safe daily living (*World Health Organization, 2022*). However, during rehabilitation practice, falls occur frequently (*Pauley, Devlin & Heslin, 2006*; *Rapp et al., 2016*), prolonging hospital stays, increasing medical costs (*Morello et al., 2015*), causing physical injury, reducing the quality of life, and even causing death (*Healey et al., 2008*; *Deutschbein et al., 2023*). Falls are caused by various factors, including muscle weakness, fall history, gait and balance impairments, the use of assistive devices, poor lighting, loose carpets, and an unsafe environment in the bathroom (*American Geriatrics Society, 2001*). Among the professionals involved in healthcare services, physical and occupational therapists (rehabilitation professionals) are mainly responsible for enhancing the performance of patients' life functions (*World Confederation for Physical Therapy, 2019*; *American Occupational Therapy Association, 2021*; *World Health Organization, 2022*). Therefore, it is necessary to maintain a perspective that balances the tradeoff between the expansion of life functions and safety (*Tinetti & Kumar, 2010*).

Fall risk assessment is important for rehabilitation professionals to assess life functioning and provide appropriate support (*Williams-Roberts et al., 2021*). Rating scales, such as St. Thomas's Risk Assessment Tool in Falling Elderly Inpatients (*Oliver et al., 1997*), the Morse Fall Scale (*Morse, Morse & Tylko, 1989*), and the Downton Fall Risk Index (*Nyberg & Gustafson, 1996*), have been developed to identify those at high risk for falls (*Oliver et al., 2004*). In a review of clinical practice guidelines on fall risk assessment, the common assessment items included fall history; gait, balance, and/or mobility; medication review; vision; and environmental hazards (*Williams-Roberts et al., 2021*). In terms of fall prevention interventions, studies have reported the effectiveness of Tai Chi, the Otago

Exercise Program, medication management, vitamin D supplementation, cataract surgery, monofocal lens usage, and environmental adjustments for community-dwelling older adults (*Stevens & Lee, 2018*). In addition, a review of older adults (both community-dwelling and hospital/institutional) reported that a combination of multifactorial interventions (*e.g.*, case management, patient reminders, and staff training) may reduce the risk of falls (*Tricco et al., 2019*). Thus, although the components of fall risk assessment and its intervention strategies are evidence-based, clinical decisions regarding its use are left to the discretion of healthcare professionals (*Williams-Roberts et al., 2021*). During rehabilitation practice, interventions are required to predict fall risks and improve patients' performance of life functions while preventing falls (*Sherrington & Tiedemann, 2015*; *World Federation of Occupational Therapists , 2020*). Therefore, to provide safe and high-quality programs, rehabilitation professionals must possess the fall risk prediction ability (FRPA) necessary to make appropriate decisions about fall risk assessments and intervention strategies that are based on the characteristics of individual patients.

In Japan, Kiken Yochi Training (KYT) is widely used in the industrial field to prevent occupational accidents (*Irniza et al., 2016*). Recent studies have also applied KYT in the medical field according to professional expertise (*Hashida et al., 2017*; *Sato et al., 2017*; *Maeda et al., 2020*). Time Pressure-Kiken Yochi Training (TP-KYT) was developed by *Arihisa et al. (2019)*; it is the only assessment method that measures FRPA specific to the fall risk scenes experienced by rehabilitation professionals. The time pressure aspect of TP-KYT is adopted based on reports that experienced nurses can prevent falls in simulated patients more quickly than nursing students (*Tatsue, Tetsuji & Hiroko, 2014*). The TP-KYT illustrates five scenes that are frequently associated with falls and quantifies the degree to which fall risk can be accurately and rapidly assessed; the validity of its content has been confirmed (*Arihisa et al., 2019*). However, it is unclear which FRPA factors are measured by the TP-KYT. Clarifying the FRPA factors included in the TP-KYT can potentially help understand which perspectives should be considered for rehabilitation-related patient safety because TP-KYT was developed based on medical accidents that occur during rehabilitation practice in Japan. In addition, professional training in patient safety requires ongoing competence development that begins in the student stages and continues after certification (*JACME, 2023*). However, the content of training programs focusing on fall prevention has been noted to be inadequate, despite the need for them to be based on scientific evidence (*Shaw, Kiegaldie & Farlie, 2020*). Studies using the TP-KYT have reported that experienced clinical education groups of rehabilitation students had higher FRPA than inexperienced groups did (*Matsushita et al., 2023*). Therefore, the TP-KYT scores reflect differences in experience in clinical settings (*Arihisa et al., 2019*; *Matsushita et al., 2023*). Furthermore, FRPA was higher in professionals than in rehabilitation students (*Arihisa et al., 2019*). Rehabilitation students gain clinical experience by participating in the rehabilitation of patients under the supervision of a clinical instructor during their clinical education (*Ministry of Education, Culture, Sports, Science and Technology and Ministry of Health, Labour and Welfare, 2022*). After certification, they often implement individualized programs for inpatients on a daily basis (*Ministry of Health, Labour and Welfare, 2008*). In other words, clinical experience in rehabilitation practice with patients

differs significantly between rehabilitation professionals and students. Therefore, a model could be hypothesized in which the differences between these two groups (with or without a license) can be measured using this tool. However, this hypothesized model has not been verified. It is expected that testing the model hypothesized in this study will reveal FRPA factors that reflect differences between the two groups' clinical experience and allow for more effective fall prevention training that is based on FRPA factors planned for rehabilitation professionals and students.

Therefore, this study aims to identify the FRPA factors included in the TP-KYT and verify the model of FRPA factors related to whether participants had a rehabilitation professional license.

## MATERIALS & METHODS

### Study design

This was a cross-sectional study involving rehabilitation professionals and students.

### Participants and data collection

The participants were rehabilitation professionals working in 12 medical facilities and three nursing homes; the study included first- or fourth-year rehabilitation students (physical and occupational therapy students) from two schools (four-year universities). Participants were informed of the purpose and methods of the study and their written consent was obtained. In total, 402 rehabilitation professionals and students participated in the study. Inclusion criteria for rehabilitation professionals were therapists who were licensed physical or occupational therapists. No criteria based on years of experience were established. Second and third-year students were not recruited because the progression of basic and specialized subjects in the education curriculum differs across schools. Since the student participants in this study were affiliated with four-year universities, they were advanced learners in the rehabilitation professional field (*Japanese Association of Occupational Therapists, 2020a*). Data were collected through face-to-face and group surveys at the participants' workplaces and training schools. They were conducted in a quiet place to ensure that participants focused on the survey. The surveys were administered by occupational therapists who understood the TP-KYT procedures. This cross-sectional survey was conducted from January 2015 to April 2023. Of the 402 participants, no participants were excluded, so all participants were included in the analysis (Table 1). Of the 402 participants, a total of 176 (102 rehabilitation professionals and 74 rehabilitation students) had also participated in a previous study on a similar topic (*Arihisa et al., 2019*).

### Ethical procedure

This study was approved by Hiroshima University's Ethics Committee for Epidemiology (E-2838). Participants were informed of the study's purpose, how the data would be processed, and that they were free to abstain or withdraw from participation without any penalty. Their consent was obtained in writing.

### Measures

The study used a participants' characteristics questionnaire and the TP-KYT.

**Table 1  Participants' characteristics.**

| Characteristics | All, N = 402 | Rehabilitation Professionals, N = 184 | Rehabilitation Students, N = 218 |
|---|---|---|---|
| Age, mean (SD), years | 25.2 (7.3) | 31.4 (6.5) | 19.9 (1.9) |
| Gender, N (%) | | | |
|   Men | 210 (52.2) | 101 (54.9) | 109 (50.0) |
|   Women | 192 (47.8) | 83 (45.1) | 109 (50.0) |
| Job role, N (%) | | | |
| Rehabilitation Professionals | | | |
|   PT | 79 (42.9) | 79 (42.9) | — |
|   OT | 105 (57.1) | 105 (57.1) | — |
| Rehabilitation students | | | |
|   PT student | 86 (39.4) | — | 86 (39.4) |
|   OT student | 132 (60.6) | — | 132 (60.6) |
| Years of experience, mean (SD), years | — | 8.0 (5.8) | — |
| Year in school, N (%) | | | |
|   First-year students | — | — | 112 (51.4) |
|   Fourth-year students | — | — | 106 (48.6) |

**Notes.**

PT, physical therapist; OT, occupational therapist.

## Participants' characteristics

Age, gender, job role (physical therapist, occupational therapist, physical therapy student, or occupational therapy student), years of experience, and year in school were collected through a questionnaire.

## TP-KYT

The TP-KYT is an evaluation tool that can quantify the FRPA of rehabilitation professionals and students (*Arihisa et al., 2019*). To enhance the performance of life functions, it includes five scenes with a high frequency of medical accidents involving falls during rehabilitation practice that differ in terms of diseases, symptoms, and environments. Scene 1: Patient is sitting in bed with low arousal after cancer surgery; Scene 2: Patient with left hemiplegia is trying to shift from a wheelchair to a bed; Scene 3: Patient with cerebellar ataxia is trying to sit on a toilet seat in a bathroom; Scene 4: Patient with right hemiplegia is trying to get into the bathtub; Scene 5: Patient with lower limb muscle weakness is doing housework. In accordance with the TP-KYT implementation procedure (*Arihisa et al., 2019*), participants were given 15 s to read the instructions, and were then required to identify the potential fall risks in the illustrations within 10 s. Participants then wrote down the reasons for identifying specific areas as fall risks. This process was repeated for all five scenes.

The scoring method used weighted scores only when the description corresponded to the 24 scoring items (*e.g.*, 40 points were added if a description corresponded to an item with a score of 40, and 0 points if no description corresponded to an item with a score of 40; *Arihisa et al., 2019*). The scoring range is from 0 to 425 points. Higher scores indicate higher FRPA. The 24 scoring items and weighting scores were quantified based on the

viewpoints of rehabilitation professionals with more than 5 years of clinical experience, thus confirming their content validity (Table 2). In this study, after identifying the FRPA factors, corrected scores were calculated for each factor and used in subsequent analyses. Thus, the total score of the 24 scoring items was not used.

## Analysis procedure

First, to identify FRPA factors included in the TP-KYT, factors were extracted from the contents of the 24 scoring items constituting the TP-KYT, as presented by *Arihisa et al. (2019)* through a qualitative analysis (Table 2). The procedure involved summarizing the contents of each scoring item and categorizing them into groups, which were then classified according to their similarity and coded. The codes were then aggregated into categories (*Creswell & Plano Clark, 2010*). In this study, the FRPA factors that are measured by the TP-KYT were aggregated as categories and are henceforth referred to as factors. The first and second analysts (both occupational therapists) conducted inductive analyses separately. Disagreements were discussed with the four collaborators to reach a consensus. The six occupational therapists involved in the analysis have been licensed for more than five years and are all experts in their field (*Unsworth, 2001*). This ensured the validity of the analysis (*Creswell & Plano Clark, 2010*).

Next, to verify the model of FRPA factors based on whether participants did or did not have rehabilitation professional licenses, structural equation modeling (SEM) was conducted by setting FRPA as a latent variable and the correction scores of factors as observed variables, then drawing a path from job role to FRPA. In the analysis, correction scores were calculated for each of the aggregated factors, and the observed variables for the job role (rehabilitation professionals = 1, rehabilitation students = 0) were set and used. The correction score was calculated by dividing the factors score (acquisition score) by the factor score (full score), which is the sum of the weighted scores of the scoring items (0 to 1). This data processing was conducted to correct the full scores of the different factors, considering that the scoring items are binary in their weighting scores (*Little et al., 2013*).

The SEM was analyzed using the generalized least squares method. The goodness of fit was evaluated using root mean square error of approximation (RMSEA), goodness-of-fit index (GFI), and adjusted GFI (AGFI), using the following criteria: RMSEA<0.05, GFI>0.95, and AGFI>0.90 (*Schermelleh-Engel, Moosbrugger & Müller, 2003*). Statistical significance was defined as $p < 0.05$. Analyses were performed using IBM SPSS Statistics 28 and Amos 28 (IBM Corp., Armonk, NY, USA).

## RESULTS

### Participants' characteristics

The characteristics of the 402 participants are shown in Table 1. The participants consisted of 184 rehabilitation professionals, 101 of whom were men (54.9%), and 218 rehabilitation students, 109 of whom were men (50.0%), with mean ages of 31.4 (SD =6.5) and 19.9 (SD =1.9), respectively. The mean years of experience of the rehabilitation professionals was 8.0 (SD =5.8). Of the rehabilitation students, 112 (51.4%) were first-year students.

**Table 2   FRPA factors for the TP-KYT extracted in the qualitative analysis.**

| Factors | Codes | Scoring items | Scenes | Weighting scores | Full scores |
|---|---|---|---|---|---|
| Patient ability (PA) | The patient's lower limbs are in an inappropriate position | There is risk of falling because the sole is not on the floor | 1 | 20 | |
| | | Because the patient's left sole is not attached and there is a risk of falling and collapsing when he is standing | 2 | 30 | |
| | | Patient's foot position is inappropriate for transfer | 2 | 5 | |
| | The patient's posture is poor | The body is tilted, and there is a risk of falling down | 1 | 15 | |
| | | Because the patient is facing downward, there is a risk of falling forward and injuring the head | 1 | 5 | |
| | The patient's ability to balance is poor | Patient loses balance when seated | 3 | 5 | |
| | | The patient is putting too much weight on the cane and it may break and the patient may fall | 5 | 15 | 140 |
| | The patient's upper limbs supporting ability is poor | It is difficult to support with upper limbs due to tremor | 3 | 5 | |
| | | Both hands are occupied and cannot be used as a support for emergency | 5 | 10 | |
| | The patient's awakening level is low | The patient's awakening level is low, and there is a risk of falling | 1 | 30 | |
| Physical environment (PE) | Improper setup of welfare equipment | Wheelchair is not braked | 2 | 30 | |
| | | The angle of the L-shaped fence is open, and it is difficult to use as a support | 2 | 15 | |
| | | L-shaped fence is not fixed | 2 | 15 | |
| | Improper positioning of welfare equipment | There is a risk of falling when the patient moves because the walker is in front | 3 | 25 | |
| | | Because the position of the bath board is far away, there is a risk of falling as the patient moves | 4 | 10 | |
| | | The position of the shower chair is set to sit down from the paralyzed side | 4 | 5 | 185 |
| | Improper choice of welfare equipment | The choice of walker is inappropriate | 3 | 5 | |
| | No welfare equipment installed | There is no suitable support such as a handrail | 3 | 40 | |
| | Inappropriate clothing | The patient is wearing socks and is likely to slip | 4 | 25 | |
| | Furniture improperly positioned | Standing position is far from table | 5 | 15 | |

**Table 2** (*continued*)

| Factors | Codes | Scoring items | Scenes | Weighting scores | Full scores |
|---|---|---|---|---|---|
| | The therapist's position is far away | The therapist is positioned away from the patient and cannot respond immediately | 4 | 35 | |
| | | The therapist is far away and cannot help him immediately | 5 | 10 | |
| Human environment (HE) | No one around | There are no people around | 1 | 15 | 100 |
| | The therapist is not observing the patient | The therapist is taking notes, and cannot help immediately | 5 | 40 | |

Notes.

Concepts

Patient ability (PA): Risk of falls caused by the patients' physical and mental functions during rehabilitation practice.

Physical environment (PE): Risk of falls caused by the physical tools surrounding the patient during rehabilitation practice.

Human environment (HE): Risk of falls caused by someone who is not prepared to help if a patient falls during rehabilitation practice or by having no one near the patient.

The 24 scoring items, scenes, and weighted scores were reprinted from Arihisa et al. (2019) with permission from Wolters Kluwer Health for use under the CC-BY-NC license. The Creative Commons license does not apply to this content. Use of the material in any format is prohibited without written permission from the publisher, Wolters Kluwer Health, Inc.

**Table 3** Scores for each factor corrected between 0–1.

| | All, $N = 402$ | | Rehabilitation Professionals, $N = 184$ | | Rehabilitation Student, $N = 218$ | |
|---|---|---|---|---|---|---|
| | Mean (SD) | Range | Mean (SD) | Range | Mean (SD) | Range |
| PA | 0.34 (0.20) | 0.00–0.89 | 0.45 (0.16) | 0.00–0.89 | 0.25 (0.18) | 0.00–0.68 |
| PE | 0.41 (0.22) | 0.00–0.86 | 0.54 (0.19) | 0.00–0.86 | 0.30 (0.19) | 0.00–0.78 |
| HE | 0.34 (0.32) | 0.00–1.00 | 0.48 (0.31) | 0.00–1.00 | 0.23 (0.29) | 0.00–0.90 |

Notes.

PA, patient ability; PE, physical environment; HE, human environment.

## FRPA factors measured by the TP-KYT

By extracting 14 types of codes from 24 items, the following three FRPA factors were aggregated: (1) Patient ability (PA): risk of falls caused by patients' physical and mental functions during rehabilitation practice; (2) Physical environment (PE): risk of falls caused by the physical tools surrounding the patient during rehabilitation practice; and (3) human environment (HE): risk of falls caused by someone who is not prepared to help if a patient falls during rehabilitation practice or by having no one near the patient. Table 2 lists the codes that comprise the factors and their concepts.

## FRPA factors based on whether participants did or did not have rehabilitation professional licenses

The model analysis results are shown in Fig. 1 (Table 2 shows the full scores for PA, PE, and HE; Table 3 shows the corrected scores for PA, PE, and HE). The standardized coefficient from job role to FRPA was 0.85 ($p < 0.001$); the standardized coefficients from FRPA to the three factors were 0.58 for PA, 0.64 for PE, and 0.46 for HE ($p < 0.001$). The model demonstrated a good fit, with RMSEA<0.001, GFI =0.998, and AGFI =0.990.

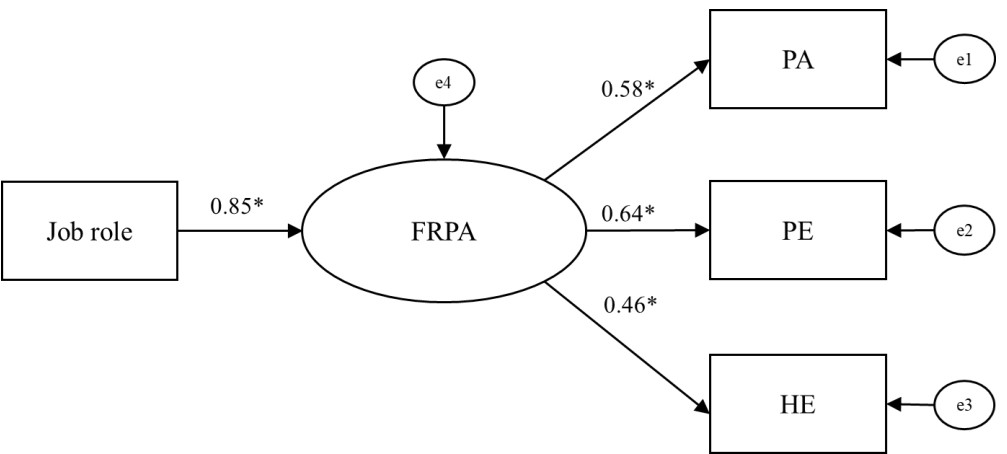

**Figure 1** **Model of FRPA factors based on whether participants did or did not have rehabilitation professional licenses.** Notes: $\chi^2$ statistic $=1.544$ ($df = 2, p = 0.462$), RMSEA $< 0.001$, GFI $=0.998$, AGFI $=0.990$. Job role: Rehabilitation professionals $=1$ ($N = 184$), Rehabilitation students $=0$ ($N = 218$). PA, PE, and HE correction scores: 0–1. Arrows indicate paths and their numerical values indicate standardization coefficients. *: $p < 0.001$. Abbreviations: FRPA, fall risk prediction ability; PA, patient ability; PE, physical environment; HE, human environment; e, error variable.

## DISCUSSION

This study revealed three FRPA factors (PA, PE, and HE) that are measured by the TP-KYT. The goodness of fit of the model consisting of the three FRPA factors based on whether participants did or did not have rehabilitation professional licenses was good.

### FRPA factors measured by the TP-KYT

PA was related to the risk of falls due to patients' physical and mental functions. In supporting patients, rehabilitation professionals work on basic, applied, and socially adaptive abilities to expand life functions, including physical and mental functions (*Ministry of Health, Labour and Welfare, 2008*). PA was extracted as a factor because rehabilitation practices need to predict the risk of falls based on PA assessment and provide quality support programs according to patient characteristics. The PE was related to the risk of falls due to the physical tools surrounding patients. As environmental adjustments by rehabilitation professionals are highly effective in preventing falls (*Stevens & Lee, 2018*), PE was identified as a factor and had the highest standardized coefficient. HE was related to the risk of falls due to medical staff not being prepared to help when the patient falls or having no one near the patient. For example, in a survey of factors contributing to the incidence of falls in hospitalized patients, it was reported that 18.4% of falls occurred because medical staff did not accurately assess the patient's fall risk while providing medical care (*Liu, Zhu & Song, 2021*). Furthermore, it has been shown that many medical accidents occur in relation to human factors, such as a lack of confirmation or inattention by medical staff (*Reason, 2000*; *Japan Council for Quality Health Care, 2012*). In this study, the codes for HE included "The therapist's position is far away" and "The therapist is not observing the patient" (Table 2).

In other words, therapist judgment and behavior may be related to the occurrence of falls, and measures focusing on human factors of rehabilitation professionals are required. Therefore, clinical practice considering HE, including rehabilitation professionals, was extracted as a factor because it leads to fall prevention. Of these three factors, while PA was similar to the assessment of gait, balance, and mobility, PE was similar to the assessment of environmental hazards, which are common components in clinical practice guidelines for fall risk assessment (*Williams-Roberts et al., 2021*). However, HE was not included in the common components of clinical practice guidelines for fall risk assessment and was a distinguishing factor of the TP-KYT. Since falls are more likely to happen when the various factors occur together (*Campbell & Robertson, 2006*), rehabilitation practices that focus on the interaction of the three factors, HE, PA, and PE, may contribute to fall prevention.

The Person–Environment–Occupation (PEO) model explains human occupational performance in terms of three aspects: person, which includes the individual's mental and physical functions and values; environment, which includes the human, physical, and social environment; and occupation, which includes tasks and engagement in life activities (*Law et al., 1996*). During rehabilitation practice, when viewed as task characteristics that enhance patients' performance of life functions, the TP-KYT's five scenes correspond to occupations. More specifically, the three factors (PA, PE, and HE) aggregated in this study are similar to the person and environment in the PEO model. Therefore, by focusing on the three factors that constitute the TP-KYT's FRPA, it may be possible to contribute to the construction of a support system that enhances the performance of life functions.

### FRPA factors based on whether participants did or did not have rehabilitation professional licenses

The results of the model analysis showed a good fit (Fig. 1). The student participants were in the process of receiving advanced education in rehabilitation. However, since the three FRPA factors were higher for professionals than for rehabilitation students, receiving professional training on fall prevention as students is necessary. In clinical settings, clinical reasoning is important for predicting and preventing behaviors that lead to patient falls (*The Japanese Society for Fall Prevention, 2023*). Clinical reasoning is classified into three domains: cognitive, psychomotor, and affective (*Huhn et al., 2019*). The cognitive domain is related to knowledge; the psychomotor domain to examination, measurement, and treatment skills, such as observation and palpation skills; and the affective domain to professionals' attitudes and emotional state (*Huhn et al., 2019*; *da Silva Araujo et al., 2022*). As the TP-KYT requires participants to read descriptions of patients' medical conditions and view illustrations that demonstrate the specific scenes to predict fall risks (*Arihisa et al., 2019*), the ability to integrate knowledge and observational skills is required. In addition, the TP-KYT score is quantified based on the risk prediction of experts with more than five years of clinical experience (*Arihisa et al., 2019*), which expresses the depth of experience in rehabilitation practice (*Arihisa et al., 2019*; *Matsushita et al., 2023*). In other words, we believe that job role and FRPA were positively related because differences in knowledge and observation of the three FRPA factors affected the depth of experience. Tests using rating scales have shown that clinical reasoning ability is higher in physical therapists than

in students (*Huhn et al., 2011*). Furthermore, the results of a literature review investigating the differences between novice and expert occupational therapists indicated that clinical reasoning is an ability that develops continuously, with progression from students to professionals (*Unsworth & Baker, 2016*). The positive association between job role and FRPA in the present study was a result shared by previous clinical reasoning studies.

Among Japanese rehabilitation professionals, about one-quarter have less than five years of clinical experience (*Japanese Association of Occupational Therapists, 2020b*; *Japanese Physical Therapy Association, 2023*). Falls that occur during rehabilitation practice are more frequent among therapists who have less than five years of experience than among those with more experience (*Maeda et al., 2018*). Providing safe, high-quality rehabilitation that avoids falls is one of the quality indicators identified by the *World Federation of Occupational Therapists (2020)*. Additionally, focusing on professional training to increase FRPA during the student years is an important patient safety measure. Continuous staff training after certification is a factor that contributes to reducing the risk of falls (*Tricco et al., 2019*). The perspective of rehabilitation professional training has the potential to effectively enhance FRPA from the student stage by including the three factors identified in the content in this study. Specifically, these findings can be used in lectures, interactive learning activities, simulation education, and so forth (*Shaw, Kiegaldie & Farlie, 2020*) to develop knowledge and observation skills that form the foundation for preventing falls in rehabilitation practice settings. Moreover, the TP-KYT can continuously measure the effects of professional training to increase FRPA, from rehabilitation students to professionals. Furthermore, the fact that HE was extracted as a characteristic factor emphasizes the importance of professional training focusing on this factor, or human factors, during rehabilitation practice. The three factors may be applied as components for rehabilitation professionals to develop assessment methods for disease-specific risk scenes for both their target patients and the diverse situations in their patients' lives. However, the factors "because the position of the bath board is far away, there is a risk of falling as the patient moves" and "the position of the shower chair is set to sit down from the paralyzed side" related to PE may need to be modified for applicability in countries where the lifestyle is different from that in Japan.

## LIMITATIONS

This study has some limitations. First, selection bias may have occurred because participants in this study may have had an interest in medical safety. Second, emotional state and stress may have influenced the results of the TP-KYT, but these were not measured in this study and should be considered in future studies. Third, the TP-KYT was designed to measure FRPA in rehabilitation medical care settings in Japan. Although common life situations in nursing and caregiving settings are used, caution should be exercised when attempting to generalize the results to settings with different cultural styles. Finally, this study aimed to verify the model of FRPA factors based on whether participants had a rehabilitation professional license but did not consider the participants' years of professional experience. Considering this as a variable may be necessary in future studies.

## CONCLUSIONS

This study revealed three FRPA factors (PA, PE, and HE) that are measured by the TP-KYT. Furthermore, the findings showed that the three FRPA factors were higher in professionals than in rehabilitation students. HE was a distinctive component, unlike common components of clinical practice guidelines for fall risk assessment. Professional training that improves FRPA needs to incorporate these three factors into the program content to provide safe, high-quality rehabilitation for patients. TP-KYT can be used as an indicator of the effectiveness of that training. Furthermore, the three factors can be utilized in developing assessment methods according to disease and the diversity of patients' lives.

## ACKNOWLEDGEMENTS

We thank the participants who took time out of their busy schedules to participate in this study. We also thank Dr. Yasushi Orihashi and Mr. Gaku Aoki (MPH), Clinical Research Center in Hiroshima, Hiroshima University Hospital, for their study design advice. In addition, we thank Dr. Hirokazu Yanagihara of Hiroshima University for lending his expertise in statistical data analysis.

### Funding

This study was supported by a JSPS KAKENHI Grant-in-Aid for Young Scientists [Grant Number 20K20249] and the Japanese Council of Senior Citizens Welfare Service [Grant Number 101]. The funders had no role in study design, data collection and analysis, decision to publish, or preparation of the manuscript.

### Grant Disclosures

The following grant information was disclosed by the authors:
A JSPS KAKENHI Grant-in-Aid for Young Scientists: 20K20249.
The Japanese Council of Senior Citizens Welfare Service: 101.

### Competing Interests

The authors declare that there are no competing interests.

### Author Contributions

- Ryohei Kishita conceived and designed the experiments, performed the experiments, analyzed the data, prepared figures and/or tables, authored or reviewed drafts of the article, and approved the final draft.
- Hideki Miyaguchi conceived and designed the experiments, performed the experiments, analyzed the data, prepared figures and/or tables, authored or reviewed drafts of the article, and approved the final draft.
- Tomoko Ohura conceived and designed the experiments, performed the experiments, analyzed the data, prepared figures and/or tables, authored or reviewed drafts of the article, and approved the final draft.

- Katsuhiko Arihisa performed the experiments, analyzed the data, authored or reviewed drafts of the article, and approved the final draft.
- Wataru Matsushita performed the experiments, analyzed the data, authored or reviewed drafts of the article, and approved the final draft.
- Chinami Ishizuki conceived and designed the experiments, performed the experiments, analyzed the data, prepared figures and/or tables, authored or reviewed drafts of the article, and approved the final draft.

### Human Ethics

The following information was supplied relating to ethical approvals (i.e., approving body and any reference numbers):

The study was conducted in accordance with the Declaration of Helsinki principles, and was approved by the Ethical Committee for Epidemiology of Hiroshima University (E-2838).

### Data Availability

The raw data are available in the Supplemental File.

### Supplemental Information

Supplemental information for this article can be found online at http://dx.doi.org/10.7717/peerj.16724#supplemental-information.

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
