# Peer review of "Fall risk prediction ability in rehabilitation professionals: structural equation modeling using time pressure test data for Kiken-Yochi Training"

_PeerJ, doi:10.7717/peerj.16724_

## Round 0.1 · original submission · Major Revisions

After rigorous reviews by three peer reviewers, I have decided to request a major revision from the authors.

Reviewer 1 ·

Basic reporting

Clear and unambiguous, professional English used throughout - English writing of this article is very good, clear and used in simple enough language that could help understanding.

Literature references, sufficient field background/context provided. - I think this is good, however some additinal information about Kiken Yochi in general and the time-senstitive kiken yochi information is lacking in introduction. Additional information maybe 2-3 sentences could provide a little more context for this tool/method.

Professional article structure, figures, tables. Raw data shared. - yes

Self-contained with relevant results to hypotheses. - yes

Experimental design

Original primary research within Aims and Scope of the journal. - Yes

Research question well defined, relevant & meaningful. It is stated how research fills an identified knowledge gap. - yes

Rigorous investigation performed to a high technical & ethical standard. - yes

Methods described with sufficient detail & information to replicate. - Mostly yes, but the scoring and correction method may need to improve its explanation

Validity of the findings

Meaningful replication encouraged where rationale & benefit to literature is clearly stated. - justification clearly provided and limitation to use has been included.

All underlying data have been provided; they are robust, statistically sound, & controlled. - yes

Conclusions are well stated, linked to original research question & limited to supporting results. - yes

Additional comments

This is a really good study focusing on fall prediction risks which will be beneficial for the discipline and will tremendously benefit patients/persons with disabilities in future. There are several implications of the findings which was clearly discussed/included in the study, among others is the need to improve on the training contents for the students to better prepare them for their professional services later.

Kiken Yochi method is interesting in safety discipline and there are limited research incorporating this method especially elsewhere from Japan. Please find several comments for improvements which are minor in nature.

1. From the abstract I read, I was unsure if this study focuses on person with disabilities/accidents, I initially thought it was for older age people, maybe because I am not from the clinical setting. May be good to improve the abstract write up to make it clear.
2. In abstract, I know there is limited space but I think it would be good to address on what is Kiken Yochi.
3. SImilarly, in introduction, a very minimal explanation has been used to explain kiken yochi, this may need to improve as to help provide context and understanding to others in the field about kiken yochi.
4. Captions of tables/diagram are not self explanatory, to improve on these.
5. Add on how this can be used to people outside Japan in Discussion.

Thank you.

Reviewer 2 ·

Basic reporting

No comment.

Experimental design

1. Please describe the 24 items and the reasons for categorisation. Why was categorisation necessary?

Validity of the findings

1. It cannot be determined that Model 1 is the optimal model as multiple models are not presented. Present the results of other models to make model comparisons.

2. Related to the comment above, I believe that the validity of the categories made by qualitative analysis can be examined by comparing models.

3.Did adding other variables to the model reduce the goodness of fit? Why did you create the model included years of experience only?

Reviewer 3 ·

Basic reporting

In clinical practice, improving the predictive ability of falls is an urgent issue. This study identified Fall risk prediction ability factors for falls in the newly developed TP-KYT. In particular, we believe that the fact that human environment (HE) was extracted is clinically significant. I commented on several issues.

introduction
Lines 65-67: It is also necessary to consider external factors of falls.
Lines 80-84: Falls prevention interventions are mentioned. Please clarify whether the intervention is targeted at the community or facility (hospital).
Discussion: HE will be the most distinctive keyword in this study. Please give further consideration regarding “The therapist’s position is far away” and “The therapist is not observing the patient” In addition, since falls are caused by a complex interplay of many factors, a discussion involving all three of these factors is also necessary.

Experimental design

abstract
Lines 37,38: Please describe the methodology about "qualitatively analyzed.”
Methods
Line 142: What is “familiar with the measurement procedures”? (regarding what?)
Lines 144,145: Because some of the data from previous studies are used, it is necessary to state in detail which data were used.
Lines 153-156: Are there any criteria for years of experience for rehabilitation professionals? If TP-KYT reflects clinical experience, shouldn't it also take into account years of experience?
Line 157-: The implementation procedures in previous studies have been described in considerable detail. Arihisa et al. are cited here and there in the sentence. I think it would be better to write it more concisely.

Validity of the findings

Results
Line 224: Is the term “tools” appropriate? I think tool is appropriate when checking Table 2, but does it not include lighting, floor surfaces, etc.? Does TP-KYT take into account external factors related to falls?

Additional comments

no comment

---

## Round 0.2 · accepted · Accept

Since most of the points were fixed in proper way, the manuscript has reached the level of publication.

Reviewer 2 ·

Basic reporting

No comment.

Experimental design

The authors responded appropriately to the points raised. Thank you.
However, when SEM, which assumes latent variables, is used in the medical and rehabilitation sciences, I consider it necessary to compare multiple models, even for hypothesis testing. When latent variables that cannot be directly observed are assumed, we consider that this has different implications from hypothesis testing in medical and rehabilitation science research. In hypothesis testing in humanities research, I have a different view.

Validity of the findings

No comment.

Additional comments

No comment.

Reviewer 3 ·

Basic reporting

Thank you for taking time to add information based on feedback, I appreciate your efforts. All previous concerns have been addressed.

Experimental design

Thank you for considering my comments.
I have no more comment for the experimental design of this paper.

Validity of the findings

Thank you for considering my comments.
I have no more comment for the validity of the findings of this paper.

Additional comments

no comment